# Carfilzomib, lenalidomide, dexamethasone, and cyclophosphamide (KRdc) as induction therapy for transplant-eligible, newly diagnosed multiple myeloma patients (Myeloma XI+): Interim analysis of an open-label randomised controlled trial

**Graham H. Jackson**[1☙*], **Charlotte Pawlyn**[2,3☙], **David A. Cairns**[4], **Ruth M. de Tute**[5], **Anna Hockaday**[4], **Corinne Collett**[4], **John R. Jones**[6], **Bhuvan Kishore**[7], **Mamta Garg**[8], **Cathy D. Williams**[9], **Kamaraj Karunanithi**[10], **Jindriska Lindsay**[11], **Alberto Rocci**[12,13], **John A. Snowden**[14], **Matthew W. Jenner**[15], **Gordon Cook**[4,16], **Nigel H. Russell**[9], **Mark T. Drayson**[17], **Walter M. Gregory**[4], **Martin F. Kaiser**[2,3], **Roger G. Owen**[5], **Faith E. Davies**[18‡], **Gareth J. Morgan**[18‡], **the UK NCRI Haemato-oncology Clinical Studies Group**

**1** Northern Institute for Cancer Research, Newcastle University, Newcastle upon Tyne, United Kingdom, **2** The Institute of Cancer Research, London, United Kingdom, **3** The Royal Marsden Hospital NHS Foundation Trust, London, United Kingdom, **4** Clinical Trials Research Unit, Leeds Institute of Clinical Trials Research, University of Leeds, Leeds, United Kingdom, **5** Haematological Malignancy Diagnostic Service, St James's University Hospital, Leeds, United Kingdom, **6** Kings College Hospital NHS Foundation Trust, London, United Kingdom, **7** University Hospitals Birmingham NHS Foundation Trust, Birmingham, United Kingdom, **8** Leicester Royal Infirmary, Leicester, United Kingdom, **9** Centre for Clinical Haematology, Nottingham University Hospital, Nottingham, United Kingdom, **10** University Hospitals of North Midlands NHS Trust, Stoke-on-Trent, United Kingdom, **11** East Kent Hospitals University NHS Foundation Trust, Canterbury, United Kingdom, **12** Manchester University NHS Foundation Trust, Manchester, United Kingdom, **13** Division of Cancer Sciences, School of Medical Sciences, Faculty of Biology, Medicine and Health, University of Manchester, Manchester, United Kingdom, **14** Royal Hallamshire Hospital, Sheffield Teaching Hospitals NHS Foundation Trust, Sheffield, United Kingdom, **15** University Hospital Southampton NHS Foundation Trust, Southampton, United Kingdom, **16** Section of Experimental Haematology, Leeds Institute of Cancer and Pathology, University of Leeds, Leeds, United Kingdom, **17** Clinical Immunology Service, Institute of Immunology and Immunotherapy, University of Birmingham, Birmingham, United Kingdom, **18** Perlmutter Cancer Center, NYU Langone Health, New York, New York, United States of America

☙ These authors contributed equally to this work.
‡ These authors are joint senior authors on this work.
* graham.jackson@newcastle.ac.uk

## Abstract

### Background

Carfilzomib is a second-generation irreversible proteasome inhibitor that is efficacious in the treatment of myeloma and carries less risk of peripheral neuropathy than first-generation proteasome inhibitors, making it more amenable to combination therapy.

### Methods and findings

The Myeloma XI+ trial recruited patients from 88 sites across the UK between 5 December 2013 and 20 April 2016. Patients with newly diagnosed multiple myeloma eligible for

**Data Availability Statement:** There are legal restrictions on sharing data that contain potentially identifying or sensitive personal information. The restrictions are imposed by The Information Commissioner's Office (https://ico.org.uk/). Data used in the current study will be made available upon request after application to the Myeloma XI data controller and the independent trial steering committee. Any requests for trial data and supporting material (data dictionary, protocol, and statistical analysis plan) should be sent to ctru-dataaccess@leeds.ac.uk. Data requestors will need to sign a data access agreement.

**Funding:** Primary financial support was from Cancer Research UK (https://www.cancerresearchuk.org/; C1298/A10410 to GJM, FED, GHJ, MTD, NR, WMG). Unrestricted educational grants from Celgene Corporation (https://www.celgene.com/; to GJM, GHJ), Amgen (https://www.amgen.com/; to GJM, GHJ) and Merck Sharp and Dohme (https://www.merck.com/; to GJM, GHJ), and funding from Myeloma UK (https://www.myeloma.org.uk/; to GJM, MFK) supported trial coordination and laboratory studies. The funders had no role in study design, data collection and analysis, decision to publish, or preparation of the manuscript.

**Competing interests:** I have read the journal's policy and the authors of this manuscript have the following competing interests: RdeT and NR declare no conflict of interest. DAC, AH and CC, report research funding from Celgene Corporation, Amgen, and Merck Sharp and Dohme. GHJ has received consultancy fees, honoraria and speakers' bureau fees from Roche, Amgen, Janssen, and Merck Sharp and Dohme; and consultancy fees, honoraria, travel support, research funding, and speakers' bureau fees from Celgene Corporation and Takeda. CP has received consultancy fees, honoraria, and travel support from Amgen, Celgene Corporation, Janssen, Sanofi and Takeda Oncology. JRJ has received honoraria and research funding from Celgene Corporation. BK has received consultancy fees, travel support, and speakers' bureau fees from Celgene Corporation, Takeda, and Janssen. MG has received travel support, research funding, and speakers' bureau fees from Janssen; travel support from Takeda; and travel support and research funding from Novartis. CDW has received honoraria, travel support, and speakers' bureau fees from Takeda, Janssen, and Celgene Corporation; honoraria and speakers' bureau fees from Amgen; and honoraria from Novartis. KK has received travel support and research funding from Celgene Corporation and Janssen. JL has received consultancy fees from

transplantation were randomly assigned to receive the combination carfilzomib, lenalidomide, dexamethasone, and cyclophosphamide (KRdc) or a triplet of lenalidomide, dexamethasone, and cyclophosphamide (Rdc) or thalidomide, dexamethasone, and cyclophosphamide (Tdc). All patients were planned to receive an autologous stem cell transplantation (ASCT) prior to a randomisation between lenalidomide maintenance and observation. Eligible patients were aged over 18 years and had symptomatic myeloma. The co-primary endpoints for the study were progression-free survival (PFS) and overall survival (OS) for KRdc versus the Tdc/Rdc control group by intention to treat. PFS, response, and safety outcomes are reported following a planned interim analysis. The trial is registered (ISRCTN49407852) and has completed recruitment. In total, 1,056 patients (median age 61 years, range 33 to 75, 39.1% female) underwent induction randomisation to KRdc ($n = 526$) or control (Tdc/Rdc, $n = 530$). After a median follow-up of 34.5 months, KRdc was associated with a significantly longer PFS than the triplet control group (hazard ratio 0.63, 95% CI 0.51–0.76). The median PFS for patients receiving KRdc is not yet estimable, versus 36.2 months for the triplet control group ($p < 0.001$). Improved PFS was consistent across subgroups of patients including those with genetically high-risk disease. At the end of induction, the percentage of patients achieving at least a very good partial response was 82.3% in the KRdc group versus 58.9% in the control group (odds ratio 4.35, 95% CI 3.19–5.94, $p < 0.001$). Minimal residual disease negativity (cutoff $4 \times 10^{-5}$ bone marrow leucocytes) was achieved in 55% of patients tested in the KRdc group at the end of induction, increasing to 75% of those tested after ASCT. The most common adverse events were haematological, with a low incidence of cardiac events. The trial continues to follow up patients to the co-primary endpoint of OS and for planned long-term follow-up analysis. Limitations of the study include a lack of blinding to treatment regimen and that the triplet control regimen did not include a proteasome inhibitor for all patients, which would be considered a current standard of care in many parts of the world.

## Conclusions

The KRdc combination was well tolerated and was associated with both an increased percentage of patients achieving at least a very good partial response and a significant PFS benefit compared to immunomodulatory-agent-based triplet therapy.

## Trial registration

ClinicalTrials.gov ISRCTN49407852.

### Author summary

#### Why was this study done?

- Although outcomes for myeloma patients have improved over recent decades, many patients will eventually relapse, and finding new treatment regimens that keep patients in first remission for longer is imperative.

Janssen; travel support from Novartis and Takeda; consultancy fees and travel support from Bristol-Myers Squibb; and honoraria and travel support from Celgene Corporation. AR has received consultancy fees from Takeda and Sanofi Genzyme, honoraria from Celgene, Novartis, Takeda, Amgen, Janssen, travel support from AbbVie, Takeda and Janssen and has been part of advisory boards for Novartis, Janssen and Amgen. JAS has received speaker fees for educational events supported by Sanofi, Janssen, Jazz, Mallinckrodt, Actelion and Gilead and is a member of a trial IDMC for Kiadis Pharma. MWJ has received consultancy fees, honoraria, travel support, and research funding from Janssen; consultancy fees, honoraria, and travel support from Takeda and Amgen; consultancy fees, honoraria, and research funding from Celgene Corporation; and consultancy fees and honoraria from Novartis. GC has received consultancy fees, honoraria, research funding, and speakers' bureau fees from Takeda, Celgene Corporation, Janssen, and Amgen; consultancy fees and honoraria from Glycomimetics and Bristol-Myers Squibb; and consultancy fees, honoraria, and speakers' bureau fees from Sanofi. MTD has equity ownership in, and is on the board of directors and advisory committee of, Abingdon Health. WMG has received consultancy fees and research funding from Celgene Corporation; research funding from Amgen and Merck Sharp and Dohme; and honoraria from Janssen. MFK has received consultancy fees and travel support from Bristol-Myers Squibb and Takeda; consultancy fees from Chugai; consultancy fees and honoraria from Janssen and Amgen; and consultancy fees, honoraria, and research funding from Celgene Corporation. RGO has received honoraria and travel support from Takeda; consultancy fees and travel support from Janssen; consultancy fees, honoraria, and research funding from Celgene Corporation. FED has received consultancy fees and honoraria from Amgen, AbbVie, Takeda, Janssen, Celgene and Roche. GJM has received research funding from Janssen; consultancy fees and honoraria from Bristol-Myers Squibb, Roche, Amgen, GSK, Karyopharm and Takeda; and consultancy fees, honoraria, and research funding from Celgene Corporation.

**Abbreviations:** ASCT, autologous stem cell transplantation; CR, complete response; CVD, cyclophosphamide, bortezomib, and dexamethasone; KRd, carfilzomib, lenalidomide, and dexamethasone; KRdc, carfilzomib, lenalidomide, dexamethasone, and cyclophosphamide; MRD, minimal residual disease; NE, not estimable; OS, overall survival;

- Treatment combinations aim to target myeloma using different mechanisms of action. This aims to prevent resistant cells remaining and leading to relapse.
- Carfilzomib has been shown to be effective in the treatment of relapsed myeloma when given in combination with lenalidomide.
- We performed this study to compare the outcomes of newly diagnosed myeloma patients treated with the combination carfilzomib, lenalidomide, dexamethasone, and cyclophosphamide (KRdc) to those of patients treated with immunomodulatory-agent-based triplet control combinations.

## What did the researchers do and find?

- A total of 1,056 patients with newly diagnosed myeloma were randomised between KRdc and control.
- Patients receiving the quadruplet KRdc combination were more likely to achieve at least a very good partial response and had longer progression-free survival than patients receiving control treatments.
- Side effects were not significantly increased with the quadruplet combination compared to control.

## What do the findings mean?

- The results suggest that KRdc can elicit deep responses and long periods of progression-free survival with tolerable side effects in newly diagnosed myeloma patients.
- Further studies are needed to compare KRdc to combinations of other new agents for the treatment of newly diagnosed myeloma patients.

## Introduction

Multiple myeloma is a malignancy of plasma cells that can lead to bone destruction, anaemia, renal dysfunction, and/or hypercalcaemia. Outcomes for patients with myeloma have improved considerably in the last few decades following the introduction of immunomodulatory agents and proteasome inhibitors, the use of autologous stem cell transplantation (ASCT), and improvements in supportive care strategies. There is a strong rationale for delivering induction treatment using combinations of agents with different mechanisms of action to overcome intra-clonal heterogeneity [1,2]. However, the additional toxicities of combinations of 3, 4, or more drugs can limit their clinical utility. Carfilzomib is a second-generation irreversible proteasome inhibitor that is structurally and mechanistically distinct from the first-generation proteasome inhibitor, bortezomib [3]. Carfilzomib has been shown to be safe and effective in relapsed multiple myeloma, both in a doublet combination with dexamethasone (Kd) in the ENDEAVOR trial [4,5] and in triplets such carfilzomib, lenalidomide, and dexamethasone (KRd), evaluated in the ASPIRE trial [6,7]. The impressive results of these studies

PFS, progression-free survival; PFS2, progression-free survival 2; Rdc, lenalidomide, dexamethasone, and cyclophosphamide; SAE, serious adverse event; Tdc, thalidomide, dexamethasone, and cyclophosphamide; VGPR, very good partial response; VRd, bortezomib, lenalidomide, and dexamethasone.

support the investigation of carfilzomib and lenalidomide given in combination for induction treatment in newly diagnosed, transplant-eligible patients. We report the results of a planned interim analysis of Myeloma XI+, a phase III randomised trial of the combination carfilzomib, lenalidomide, dexamethasone, and cyclophosphamide (KRdc) in comparison to an immuno-modulatory-agent-based triplet control for newly diagnosed multiple myeloma patients destined for ASCT.

## Methods

### Study design and participants

The Myeloma XI+ trial is a multi-centre, randomised, open-label, phase III trial. This planned interim analysis reports the first of 2 co-primary outcomes—progression-free survival (PFS)—and secondary and safety outcomes. The study is closed for accrual, but follow-up continues for planned long-term analysis. All authors contributed to the development of the manuscript, approved the final version, and vouch for the accuracy and completeness of the data and for the fidelity of the trial to the protocol. A completed CONSORT checklist is available (Text A in S1 Text). The study is registered with the ISRCTN registry (ISRCTN49407852) and EU Clinical Trials Register (2009-010956-93).

Eligible patients were recruited from 88 National Health Service sites across the UK (Table A in S1 Text) between 5 December 2013 and 20 April 2016. These sites ranged from academic medical centres to local district general hospitals. Inclusion criteria allowed the enrolment of patients who were 18 years or older and had newly diagnosed symptomatic myeloma. Patients were excluded if they had previous or concurrent active malignancies (including myelodysplastic syndrome), peripheral neuropathy of grade 2 or greater, acute renal failure (characterised by creatinine > 500 μmol/l, urine output < 400 ml/day, or requirement for dialysis and unresponsive to up to 72 hours of rehydration), or active or prior hepatitis C virus infection. Further details regarding the inclusion and exclusion criteria can be found in the protocol (Text B in S1 Text).

Myeloma XI+ followed the Myeloma XI trial using a seamless adaptation to implement a novel treatment approach soon after it became available. Myeloma XI had pathways for both transplant-eligible and -ineligible patients. Patients in the transplant-eligible pathway were randomised between thalidomide, dexamethasone, and cyclophosphamide (Tdc) and lenalidomide, dexamethasone, and cyclophosphamide (Rdc), with subsequent response-adapted intensification and maintenance randomisations that were also carried forward into Myeloma XI+. Myeloma XI+ was only for transplant-eligible patients and was designed and opened before Myeloma XI data had matured; as such, the primary outcome was the comparison between KRdc and the triplet approaches studied in Myeloma XI and continued into Myeloma XI+. Results of the Tdc versus Rdc randomisation in transplant-eligible patients from Myeloma XI showed a small, but statistically significant improvement in PFS and overall survival (OS) associated with receiving Rdc [8]; results of the cyclophosphamide, bortezomib, and dexamethasone (CVD) intensification randomisation and the maintenance randomisation are also published [9,10]. This paper reports only patients contemporaneously randomised between KRdc and Rdc/Tdc as per the statistical analysis plan for the study.

### Randomisation and treatment

Patients were randomly assigned in a 2:1:1 ratio between KRdc, Rdc, and Tdc (Fig 1). All randomisations were performed at the Clinical Trials Research Unit (Leeds, UK) using a centralised, automated 24-hour telephone system. Due to the nature of the intervention, patients and their physicians were aware of the treatment allocation. A minimisation algorithm with a

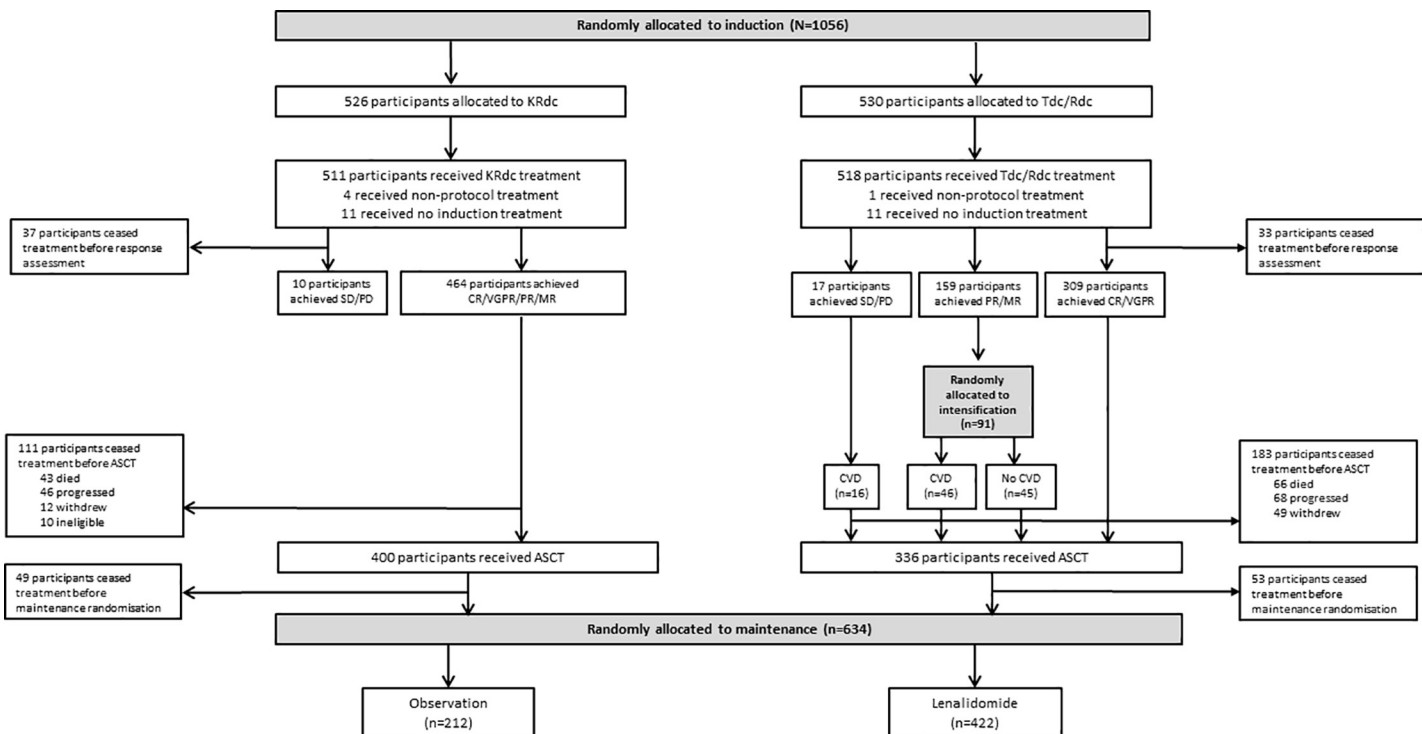

**Fig 1. Patient disposition.** ASCT, autologous stem cell transplantation; CR, complete response; CVD, cyclophosphamide, bortezomib, and dexamethasone; KRdc, carfilzomib, lenalidomide, dexamethasone, and cyclophosphamide; MR, minimal response; PD, progressive disease; PR, partial response; Rdc, lenalidomide, dexamethasone, and cyclophosphamide; SD, stable disease; Tdc, thalidomide, dexamethasone, and cyclophosphamide; VGPR, very good partial response.

random element was used to avoid chance imbalances in 6 variables measured at trial entry: β2 microglobulin (<3.5 mg/l versus 3.5 to <5.5 mg/l versus ≥5.5 mg/l versus or unknown), haemoglobin (<115 g/l versus ≥115 g/l for men; <95 g/l versus ≥95 g/l for women), corrected serum calcium (<2.6 mmol/l versus ≥2.6 mmol/l), serum creatinine (<140 μmol/l versus ≥140 μmol/l), platelets (<150 × $10^9$ cells/l versus ≥150 × 109 cells/l), and centre.

Initial induction treatment with KRdc, Rdc, or Tdc was administered, in the absence of toxicity, consent withdrawal, or progression, for a minimum of 4 cycles and to maximum response. Patients in the KRdc group received 28-day cycles of carfilzomib 36 mg/m$^2$ IV on days 1, 2, 8, 9, 15, and 16 (20 mg/m$^2$ on days 1 and 2); lenalidomide 25 mg PO on days 1–21; dexamethasone 40 mg PO on days 1–4, 8, 9, 15, and 16; and cyclophosphamide 500 mg PO on days 1 and 8. In cycle 1 intravenous hydration was recommended prior to administration of carfilzomib with 250 to 500 ml of normal saline or other appropriate IV fluid. Carfilzomib was dose capped at a body surface area of 2.2 m$^2$. Investigators were advised to routinely evaluate and treat hypertension and monitor patients for cardiac events.

Patients in the Rdc group received 28-day cycles of lenalidomide 25 mg PO on days 1–21; dexamethasone 40 mg PO on days 1–4 and 12–15; and cyclophosphamide 500 mg PO on days 1 and 8. Patients in the Tdc group received 21-day cycles of thalidomide 100 mg (increasing to 200 mg as tolerated) PO on days 1–21; dexamethasone 40 mg PO on days 1–4 and 12–15; and cyclophosphamide 500 mg PO on days 1, 8, and 15. Patients who received Tdc/Rdc underwent response-adapted intensification, with those achieving only minimal or partial response undergoing a randomisation to a proteasome inhibitor (bortezomib)–containing triplet (CVD) or no further therapy. All patients who received Tdc/Rdc with stable or progressive disease received CVD. CVD was administered in 21-day cycles of cyclophosphamide 500 mg PO

on days 1 and 8; bortezomib 1.3 mg/m$^2$ SC or IV on days 1, 4, 8, and 11; and dexamethasone 20 mg PO on days 1, 2, 4, 5, 8, 9, 11, and 12.

Peripheral blood stem cell harvest commenced after the patient had completed induction therapy, followed by high-dose melphalan and stem cell return according to local practice but with the intention to deliver melphalan 200 g/m$^2$ (except 140 mg/m$^2$ in those with renal insufficiency, defined as serum creatinine $\geq$ 200 μmol/l prior to transplant). A maintenance randomisation at 3 months post-ASCT compared lenalidomide (10 mg PO on days 1–21 of a 28-day cycle until disease progression) to observation.

## Endpoints and assessments

The co-primary endpoints of the trial were PFS and OS. PFS was defined as the time from randomisation to progressive disease or death from any cause. OS was defined as the time from randomisation to death from any cause or last follow-up. Secondary efficacy endpoints included the percentage of patients in remission (very good partial response [VGPR] or complete response [CR]), overall response, toxicity, and progression-free survival 2 (PFS2), defined as the time from randomisation to the date of second progressive disease, start of third anti-myeloma treatment, or death from any cause.

The therapeutic efficacy was evaluated in the molecular risk subgroups of standard risk, high risk (defined as 1 adverse molecular abnormality), and ultra-high risk (2 or more adverse molecular abnormalities) and was pre-specified by protocol. Adverse molecular abnormalities were defined as gain(1q), del(17p), t(4;14), t(14;16), or t(14;20). Molecular risk profiling for the majority of patients (339/383) was performed centrally using multiplex ligation-dependent probe amplification (MLPA) and quantitative real-time PCR (qRT-PCR) on DNA/RNA extracted from CD138-selected plasma cells from bone marrow biopsies of patients taken prior to treatment commencing. In this approach, qRT-PCR is used to assay the expression of translocation gene partners including t(4;14) MMSET and FGFR3, t(14;16) MAF, and t(14;20) MAFB. MLPA was used to assay copy number by including probesets at sites of the commonly deleted and amplified regions in myeloma, e.g., at genes *CKS1B* on 1q21.3 and *TP53* on 17p13. These techniques have been previously validated and provide equivalent results to interphase fluorescence in situ hybridisation (iFISH) [11–13]. For the remaining patients (44/383), local cytogenetic (FISH) reports were centrally reviewed, and only those with a valid result for all risk markers were included in this analysis.

The presence of minimal residual disease (MRD) was assessed using a validated flow cytometry assay (sensitivity $\leq$ 0.004%, cutoff $4 \times 10^{-5}$ bone marrow leucocytes) performed at a single central laboratory on bone marrow aspirates obtained at the end of initial induction and at 100 days after ASCT. A minimum of 500,000 cells were evaluated with 8-color antibody combinations including CD138, CD38, CD45, CD19, CD56, CD27, CD81, and CD117.

Paraprotein (Sebia, France) and serum free light chains (Freelite, The Binding Site, Birmingham, UK) were assessed at least every 2 months for the first 2 years and then at least every 3 months until disease progression. Urine light chain excretion (Sebia, France) was assessed to confirm CR. Response and progressive disease were assessed on the basis of IMWG uniform response criteria [14,15] and reviewed centrally by an expert panel masked to treatment allocation. Adverse events were graded according to the US National Cancer Institute Common Terminology Criteria for Adverse Events (NCI CTCAE), version 4.0. Adverse reactions were assessed at the start of each treatment cycle. Serious adverse events (SAEs) were reported for all patients from the date of randomisation until 30 days after the date of disease progression except in the case of serious adverse reactions or second primary malignancies, which were collected for the duration of the trial.

## Statistical analysis

The primary analysis was performed in the intention-to-treat population, which included all patients who underwent randomisation. The safety population included all patients who received at least 1 dose of the trial treatment. The primary analysis was designed to compare KRdc to the triplet control group, which consisted of patients randomised to receive either Rdc or Tdc. Exploratory analysis was specified to compare KRdc with whichever of the triplets was superior. The primary endpoint of PFS was compared between the treatment groups with the use of a Cox regression model adjusting for the stratification factors. The adjusted treatment effect (hazard ratio) and corresponding 95% confidence interval were estimated with the use of this model. Other time-to-event efficacy endpoints were analysed similarly. The proportional hazards assumption was assessed informally by plotting the hazards over time (i.e., the log cumulative hazard plot) for each treatment group, and formally using the simulation-based method of Lin et al. [16]. Subgroup analysis was pre-specified for the presence or absence of individual adverse cytogenetic abnormalities and cytogenetic risk status. We did a likelihood ratio test for heterogeneity of treatment effect using Cox models identical to those used for the main analysis, with the inclusion of terms for the subgroup in question and the appropriate interaction term. The reported test for heterogeneity for subgroup analysis corresponds to a 1-degree-of-freedom test for 2 category subgroups and a 2-degrees-of-freedom test for 3 category subgroups. Continuous variables were summarised with the use of descriptive statistics, and categorical variables were summarised as numbers and percentages. Time-to-event variables were summarised with the use of the Kaplan–Meier method. We analysed binary endpoints using a logistic regression model adjusting for the stratification factors.

Of 2 planned interim analyses, the first, reported here, evaluated PFS when at least 50% of participants had experienced progressive disease or death. To ensure that an overall significance level of 5% was maintained, we used the O'Brien and Fleming alpha-spending function [17] (interim analysis bound 0.5%, final analysis bound 4.7%). The bound for the interim analysis was advisory, with the decision to release results at the recommendation of the Independent Myeloma XI/+ Data Monitoring and Ethics Committee (DMEC) and the Independent Myeloma XI/+ Trial Steering Committee (TSC). The interim analysis was done and presented to the DMEC on 12 June 2018. The DMEC recommended to the TSC that the data be released, and the TSC ratified this decision on 21 June 2018. The final PFS analysis is planned when 703 progression or death events have been reported, and the final OS analysis when 466 deaths have been reported. We estimated that a sample size of 1,044 patients would provide the trial with 80% power to detect a risk of disease progression or death that was 19% lower, and also 80% power to detect a risk of death that was 23% lower, with KRdc compared to the triplet control group. This is under the assumption of exponential survival times, with 2 years of recruitment and 4 years of follow-up, at a 2-sided alpha level of 0.05. All reported $p$-values are 2-sided and considered significant at an overall significance level of 5%.

## Ethics statement

The study was approved by the national ethics review board (National Research Ethics Service, London, UK), the institutional review boards of the participating centres, and the competent regulatory authority (Medicines and Healthcare Products Regulatory Agency, London, UK), and was undertaken according to the Declaration of Helsinki and the principles of Good Clinical Practice as espoused in the UK Medicines for Human Use (Clinical Trials) Regulations 2004. All patients provided written informed consent.

## Results

Patients were enrolled between December 2013 and April 2016 at 88 sites in the UK. Overall, 1,056 patients underwent induction randomisation (Fig 1): 526 were allocated to KRdc and 530 to the control group (265 to Rdc and 265 to Tdc). The median follow-up for this analysis is 34.5 months (IQR 27.9 to 41.3), and at the time of analysis all patients had completed their induction therapy. The groups were well matched across baseline variables, with median age 61 years (range 33–75) (Table 1).

### Efficacy

Disease progression or death occurred in 411 patients (171 of 526 patients [32.5%; 101/526, 19.2%, progression and 70/526, 13.3%, death] in the KRdc group and 240 of 530 patients [45.3%; 151/530, 28.5%, progression and 89/530, 16.8%, death] in the control group). The median PFS is not yet estimable in the KRdc group, compared to a median of 36.2 months in the control group. The hazard ratio for disease progression or death was 0.63 (95% CI 0.51 to 0.76, $p < 0.001$), corresponding to a 37% lower risk in the KRdc group (Fig 2A). There was no evidence of violation of the assumption of proportional hazards on inspection or using the method of Lin et al. ($p = 0.379$). The proportion of patients with 3-year PFS estimated by the Kaplan–Meier method was 64.5% (95% CI 59.9% to 69.1%) in the KRdc group and 50.3% (95% CI 45.4% to 55.3%) in the control group.

The improvement in PFS was seen across all subgroups defined by baseline characteristics, with no significant heterogeneity detected except with respect to age (Fig 2B). For transplant-eligible patients of all ages, KRdc was associated with significantly prolonged PFS compared to the control group, but those aged over 65 appeared to benefit to a greater extent, with a hazard ratio of 0.45 (95% CI 0.31 to 0.65) compared to those 65 or under, where a hazard ratio of 0.70 (95% CI 0.55 to 0.90) was seen ($p_{het} = 0.035$). Improved PFS was seen in all molecular risk groups, with no significant heterogeneity between groups (standard risk: hazard ratio 0.62, 95% CI 0.39 to 0.98, median PFS KRdc not estimable [NR] versus control 37 months; high risk: 0.68, 95% CI 0.40 to 1.14, median PFS KRdc NR versus control 37 months; ultra-high risk: 0.50, 95% CI 0.20 to 1.25, median PFS KRdc 36 months versus control 20 months; $p_{het} = 0.7841$) (Fig 2B and Fig A in S1 Text). The results were consistent within each of the groups with the individual risk lesions t(4;14) and del(17p) (Fig 2B and Fig B in S1 Text). An exploratory analysis to compare all 3 induction regimens demonstrates prolonged PFS associated with KRdc compared to both Rdc and Tdc induction when analysed separately (Fig 2C).

Deeper response rates were seen at the end of induction in patients treated with KRdc compared to the control group, with an odds ratio for achieving at least VGPR of 4.35 (95% CI 3.19–5.94, $p < 0.001$). At the end of induction, 82.3% of patients in the KRdc group achieved VGPR or better compared to 58.9% in the control group (Table 2). A higher proportion of patients in the KRdc group (74.9%) were able to undergo ASCT compared to those in the control group (63.8%). Post-ASCT VGPR or better was achieved in 91.9% of patients in the KRdc group compared to 79.3% in the control group.

At the end of induction, MRD negativity (MRD−) was achieved in more patients in the KRdc group (55% [90/164]) than in the control group (17% [26/157]; 11% [8/74] with Tdc, 22% [18/83] with Rdc). PFS for MRD− patients was improved compared to MRD+ patients (percent PFS at 3 years from MRD assessment, MRD− 78.7% versus MRD+ 51.9%). Interestingly, there were differences in outcome within the MRD− group, with patients receiving KRdc doing better (percent PFS at 3 years from MRD assessment, KRdc MRD− 81.8% versus control MRD− 68.1% [Tdc 62.5% and Rdc 70.6%]). In the MRD+ group the outcomes were similar (percent PFS at 3 years from MRD assessment, KRdc MRD+ 49.9% versus control

**Table 1. Baseline characteristics.**

| Characteristic | Primary endpoint comparison | | Rdc (n = 265) | Tdc (n = 265) |
|---|---|---|---|---|
| | KRdc (n = 526) | Control (Tdc/Rdc combined) (n = 530) | | |
| Median age (range)—years | 61 (33 to 75) | 62 (36 to 74) | 62 (36 to 74) | 61 (38 to 74) |
| Age group—n (%) | | | | |
| ≤65 years | 385 (73.2%) | 370 (69.8%) | 179 (67.5%) | 191 (72.1%) |
| >65 years | 141 (26.8%) | 160 (30.2%) | 86 (32.5%) | 74 (27.9%) |
| >70 years | 12 (2.3%) | 24 (4.5%) | 12 (4.5%) | 12 (4.5%) |
| Sex—n (%) | | | | |
| Male | 317 (60.3%) | 326 (61.5%) | 170 (64.2%) | 156 (58.9%) |
| Female | 209 (39.7%) | 204 (38.5%) | 95 (35.8%) | 109 (41.1%) |
| Ethnicity—n (%) | | | | |
| White | 494 (93.9%) | 482 (90.9%) | 243 (91.7%) | 239 (90.2%) |
| Black (Black Caribbean, Black African, other) | 9 (1.7%) | 6 (1.1%) | 5 (1.9) | 1 (0.4%) |
| Asian (Indian, Pakistani, Bangladeshi, other) | 9 (1.7%) | 17 (3.2%) | 7 (2.6%) | 10 (3.87%) |
| Other | 8 (1.5%) | 2 (0.4%) | 1 (0.4%) | 1 (0.4%) |
| Unknown | 6 (1.1%) | 23 (4.3%) | 9 (3.4%) | 14 (5.3%) |
| WHO performance status—n (%) | | | | |
| 0 | 225 (42.8%) | 224 (42.3%) | 101 (38.1%) | 123 (46.4%) |
| 1 | 194 (36.9%) | 182 (34.3%) | 98 (37.0%) | 86 (32.5%) |
| 2 | 58 (11.0%) | 63 (11.9%) | 30 (11.3%) | 33 (12.5%) |
| 3 | 20 (3.8%) | 26 (4.9%) | 17 (6.4%) | 9 (3.4%) |
| 4 | 1 (0.2%) | 2 (0.4%) | 1 (0.4%) | 1 (0.4%) |
| Unknown | 28 (5.3%) | 31 (5.8%) | 18 (6.8%) | 13 (4.9%) |
| ISS stage—n (%) | | | | |
| I | 167 (31.7%) | 164 (30.9%) | 80 (30.2%) | 84 (31.7%) |
| II | 198 (37.6%) | 194 (36.6%) | 95 (35.8%) | 99 (37.4%) |
| III | 117 (22.2%) | 122 (23.0%) | 63 (23.8%) | 59 (22.3%) |
| Unknown | 44 (8.4%) | 50 (9.4%) | 27 (10.2%) | 23 (8.7%) |
| Immunoglobulin subtype—n (%) | | | | |
| IgG | 299 (56.8) | 345 (65.1%) | 173 (65.3%) | 172 (64.9%) |
| IgA | 131 (24.9%) | 108 (20.4%) | 49 (18.5%) | 59 (22.3%) |
| IgM | 4 (0.8%) | 3 (0.6%) | 1 (0.4%) | 2 (0.8%) |
| IgD | 5 (1.0%) | 4 (0.8%) | 2 (0.8%) | 2 (0.8%) |
| Light chain only | 81 (15.4%) | 66 (12.5%) | 38 (14.3%) | 28 (10.6%) |
| Non-secretor | 4 (0.8%) | 4 (0.8%) | 2 (0.8%) | 2 (0.8%) |
| Unknown | 2 (0.4%) | 0 | 0 | 0 |
| Median creatinine (range)—μmol/l | 82 (40 to 395) | 81 (30 to 649) | 82 (36 to 446) | 80 (30 to 649) |
| Unknown—n (%) | 2 (0.4%) | 2 (0.4%) | 1 (0.4%) | 1 (0.4%) |
| Median LDH (range)—IU/l | 248 (83 to 1,510) | 257 (2 to 1,477) | 257 (5 to 1,477) | 256 (2 to 774) |
| Unknown—n (%) | 113 (21.5%) | 125 (23.6%) | 66 (24.9%) | 59 (22.3%) |
| Molecular risk assessment available—n (%) | 204 (38.8%) | 179 (33.8%) | 85 (32.1%) | 94 (35.5%) |
| Molecular risk—n (% of those available) | | | | |
| Standard | 101 (49.5%) | 103 (57.5%) | 47 (55.3%) | 56 (59.6%) |
| High risk[†] | 81 (39.7%) | 60 (33.5%) | 31 (36.5%) | 29 (30.9%) |
| Ultra-high risk[†] | 22 (10.8%) | 16 (8.9%) | 7 (8.2%) | 9 (9.6%) |
| Molecular risk lesions—n (% of those available) | | | | |
| t(4;14) | 28 (13.7%) | 22 (11.2%) | 11 (12.9%) | 11 (11.7%) |

*(Continued)*

**Table 1.** (Continued)

| Characteristic | Primary endpoint comparison | | Rdc (*n* = 265) | Tdc (*n* = 265) |
|---|---|---|---|---|
| | KRdc (*n* = 526) | Control (Tdc/Rdc combined) (*n* = 530) | | |
| t(14;16) | 7 (3.4%) | 0 (0%) | 0 (0%) | 0 (0%) |
| t(14;20) | 2 (1.0%) | 0 (0%) | 0 (0%) | 0 (0%) |
| del(17p) | 17 (8.3%) | 11 (6.1%) | 4 (4.7%) | 7 (7.4%) |
| gain(1q) | 71 (34.8%) | 60 (33.5%) | 30 (35.3%) | 30 (31.9%) |

[†]High-risk molecular abnormalities were defined as gain(1q), t(4;14), t(14;16), t(14;20), and del(17p). Ultra-high risk was defined as the presence of more than 1 high-risk lesion.

ISS, International Staging System; KRdc, carfilzomib, lenalidomide, dexamethasone, and cyclophosphamide; LDH, lactate dehydrogenase; Rdc, lenalidomide, dexamethasone, and cyclophosphamide; Tdc, thalidomide, dexamethasone, and cyclophosphamide.

MRD+ 52.8% [Tdc 61.8% and Rdc 40.7%]). At 100 days after ASCT, the rate of MRD− had improved in all groups, but remained higher in the KRdc group (75% [152/202]) than in the control group (50% [80/160]; 51% [40/79] with Tdc, 49% [40/81] with Rdc). The achievement of MRD− status after ASCT was also associated with improved outcomes.

OS data were immature at the time of analysis, with follow-up continuing. Data for PFS2, a key secondary endpoint, showed a significantly improved outcome for patients receiving KRdc. Second disease progression or death had occurred in 191 patients (85 of 526 patients [16.2%] in the KRdc group and 106 of 530 patients [20.0%] in the control group). The proportion of patients with 3-year PFS2 was 81.8% (95% CI 78.0% to 85.6%) in the KRdc group compared to 75.1% (95% CI 70.7% to 79.6%) in the control group. (Fig 3A). The hazard ratio for disease progression or death was 0.75 (95% CI 0.56 to 0.99, $p$ = 0.0451), corresponding to a 25% lower risk in the KRdc group compared to the Tdc/Rdc control group. An exploratory analysis to compare all 3 induction regimens demonstrates prolonged PFS2 associated with KRdc compared to Tdc but not Rdc, when analysed separately (Fig 3B).

## Safety

Induction treatment was planned for a minimum of 4 cycles and to continue to maximum response. The median number of treatment cycles received was 4 (range 1 to 12) in the KRdc group and 6 (1 to 15) in the control group (Rdc median 5, Tdc median 6). The use of a dose modification schedule in the event of adverse events allowed the vast majority of patients to complete at least 4 cycles of induction therapy. Patients not completing 4 cycles numbered 50 (9.5%) in the KRdc group and 43 (8.1%) in the Tdc/Rdc control group (Rdc 20, Tdc 23). The most common reasons cited for stopping induction therapy early were clinician choice (KRdc 22 patients, Tdc/Rdc control 26 patients [Rdc 12, Tdc 14]) or unacceptable toxicity (KRdc 20 patients, Tdc/Rdc control 14 patients [Rdc 4, Tdc 11]); more than 1 reason could be cited per withdrawal. Overall, the majority of patients stopped induction therapy because they had reached maximum response as per protocol. For patients completing 4 or more cycles, maximum response was cited as the reason for stopping in 430/526 (81.7%) in the KRdc group and 413/530 (77.9%) in the Tdc/Rdc control group (Rdc 216, Tdc 197). Clinician choice was cited in 37/526 (7.0%) in the KRdc group and 61/530 (11.5%) in the Tdc/Rdc control group (Rdc 24, Tdc 37). Unacceptable toxicity was cited in 25/526 (4.8%) in the KRdc group and 26/530 (4.9%) in the Tdc/Rdc control group (Rdc 7, Tdc 19).

Adverse reactions of any grade that were reported in more than 10% of patients in any treatment group, grade 3–4 reactions reported in more than 5% of patients in any treatment

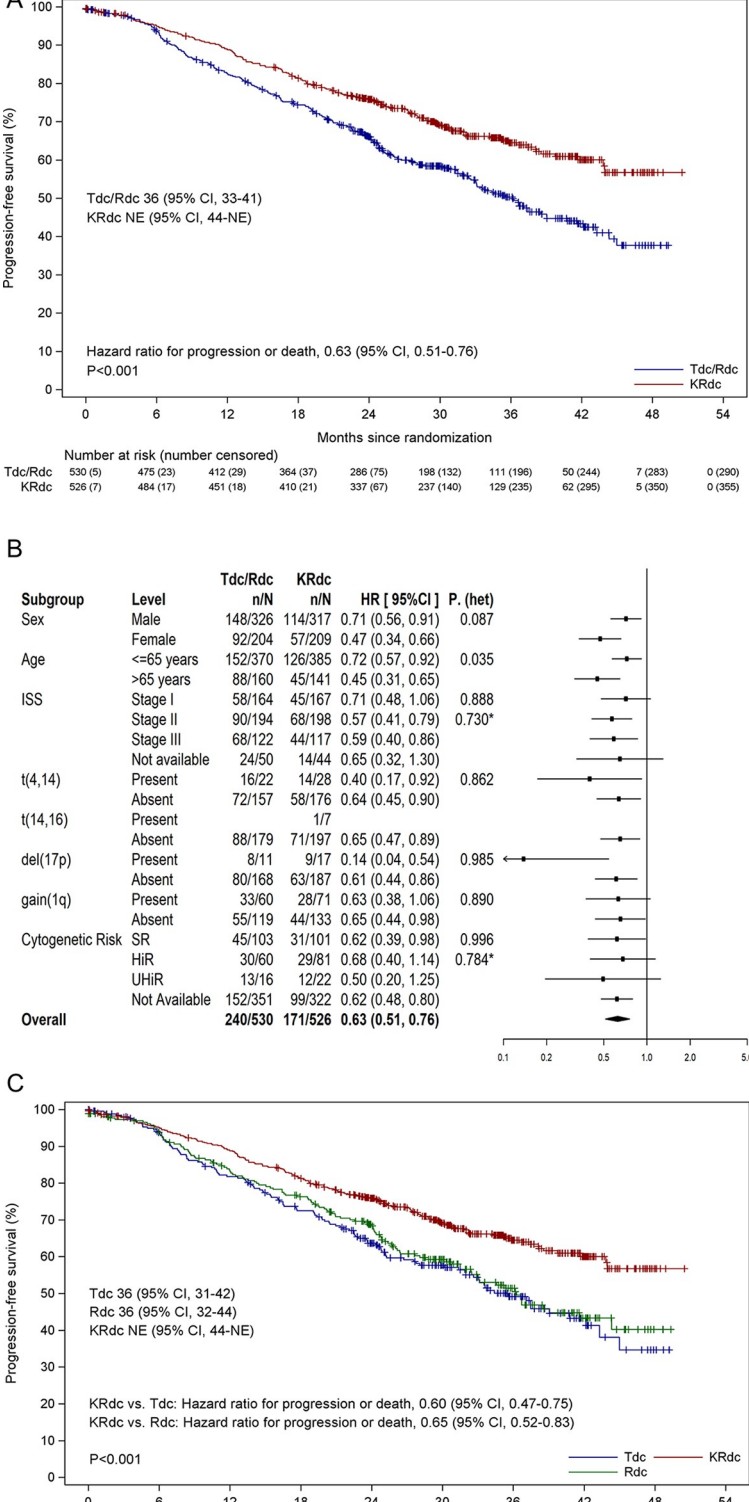

**Fig 2. Progression-free survival.** (A) Progression-free survival for KRdc compared to the Tdc/Rdc control group. (B) Subgroup analysis of progression-free survival with KRdc compared to the Tdc/Rdc control group. *Performed only

for patients with subgroup data available. (C) Progression-free survival for KRdc compared to the Tdc and Rdc groups separately. HiR, high risk; ISS, International Staging System; KRdc, carfilzomib, lenalidomide, dexamethasone, and cyclophosphamide; NE, not estimable; P. (het), *p*-value from likelihood ratio test for heterogeneity of effect; Rdc, lenalidomide, dexamethasone, and cyclophosphamide; SR, standard risk; Tdc, thalidomide, dexamethasone, and cyclophosphamide; UHiR, ultra-high risk.

group, and all grade 5 reactions are shown in Table 3. The most common adverse reactions were haematological. Grade 3 or 4 neutropenia occurred in 16.4% KRdc, 22.3% Rdc, and 12.8% Tdc patients; anaemia in 10.2% KRdc, 5.8% Rdc, and 4.7% Tdc patients; and thrombocytopenia in 8.4% KRdc, 2.3% Rdc, and 1.2% Tdc patients. There was no apparent increase in

**Table 2. Response rates.**

| Timepoint and response | Primary comparison | | Rdc | Tdc |
|---|---|---|---|---|
| | KRdc | Control (Tdc/Rdc) | | |
| **At end of initial induction** | *n* = 526 | *n* = 530 | *n* = 265 | *n* = 265 |
| CR | 93 (17.7%) | 37 (7.0%) | 19 (7.1%) | 18 (6.8%) |
| nCR | 203 (38.6) | 142 (26.8%) | 90 (34.0) | 52 (19.6%) |
| VGPR | 137 (26.0%) | 133 (25.1%) | 63 (23.8%) | 70 (26.4%) |
| ≥VGPR | 433 (82.3%) | 312 (58.9%) | 172 (64.9%) | 140 (52.8%) |
| PR | 43 (8.2%) | 154 (29.1%) | 66 (24.9%) | 88 (33.2%) |
| MR | 3 (0.6%) | 18 (3.4%) | 7 (2.6%) | 11 (4.2%) |
| SD | 1 (0.2%) | 3 (0.6%) | 0 (0.0%) | 3 (1.1%) |
| PD | 6 (1.1%) | 12 (2.3%) | 5 (1.9%) | 7 (2.6%) |
| Early death* | 6 (1.1%) | 5 (0.9%) | 2 (0.8%) | 3 (1.1%) |
| Missing | 34 (6.7%) | 26 (4.9%) | 13 (4.9%) | 13 (4.9%) |
| **At 100 days post-ASCT$** | *n* = 394 | *n* = 338 | *n* = 179 | *n* = 159 |
| CR | 122 (31.0%) | 81 (24.0%) | 41 (22.9%) | 40 (25.2%) |
| nCR | 152 (38.6%) | 107 (31.7%) | 60 (33.5%) | 47 (29.6%) |
| VGPR | 88 (22.3%) | 80 (23.7%) | 46 (25.7%) | 34 (21.4%) |
| ≥VGPR | 362 (91.9%) | 268 (79.3%) | 147 (82.1%) | 121 (76.1%) |
| PR | 23 (5.8%) | 54 (16.0%) | 26 (14.5%) | 28 (17.9%) |
| MR | 0 (0.0%) | 0 (0.0%) | 0 (0.0%) | 0 (0.0%) |
| SD | 0 (0.0%) | 0 (0.0%) | 0 (0.0%) | 0 (0.0%) |
| PD | 2 (0.5%) | 6 (1.8%) | 4 (2.2%) | 2 (1.3%) |
| Early death# | 1 (0.3%) | 1 (0.6%) | 0 (0.0%) | 1 (0.6%) |
| Missing | 6 (1.5%) | 9 (2.7%) | 2 (1.1%) | 7 (4.4%) |
| MRD-negative status | | | | |
| **At end of initial induction** | *n* = 164 | *n* = 157 | *n* = 83 | *n* = 74 |
| MRD negative | 90 (54.9%) | 20 (12.7%) | 18 (21.7%) | 8 (10.8%) |
| **At day 100 after ASCT** | *n* = 202 | *n* = 160 | *n* = 79 | *n* = 81 |
| MRD negative | 152 (75.2%) | 80 (50.0%) | 40 (50.6%) | 40 (49.4%) |

Data given as *n* (percent).

*All-cause death within 60 days of randomisation.

$Reported of these undergoing ASCT.

#All-cause death within 100 days of high-dose melphalan dose.

ASCT, autologous stem cell transplantation; CR, complete response; KRdc, carfilzomib, lenalidomide, dexamethasone, and cyclophosphamide; MR, minimal response; MRD, minimal residual disease; nCR, complete response without bone marrow confirmation; PD, progressive disease; PR, partial response; SD, stable disease; Rdc, lenalidomide, dexamethasone, and cyclophosphamide; Tdc, thalidomide, dexamethasone, and cyclophosphamide; VGPR, very good partial response.

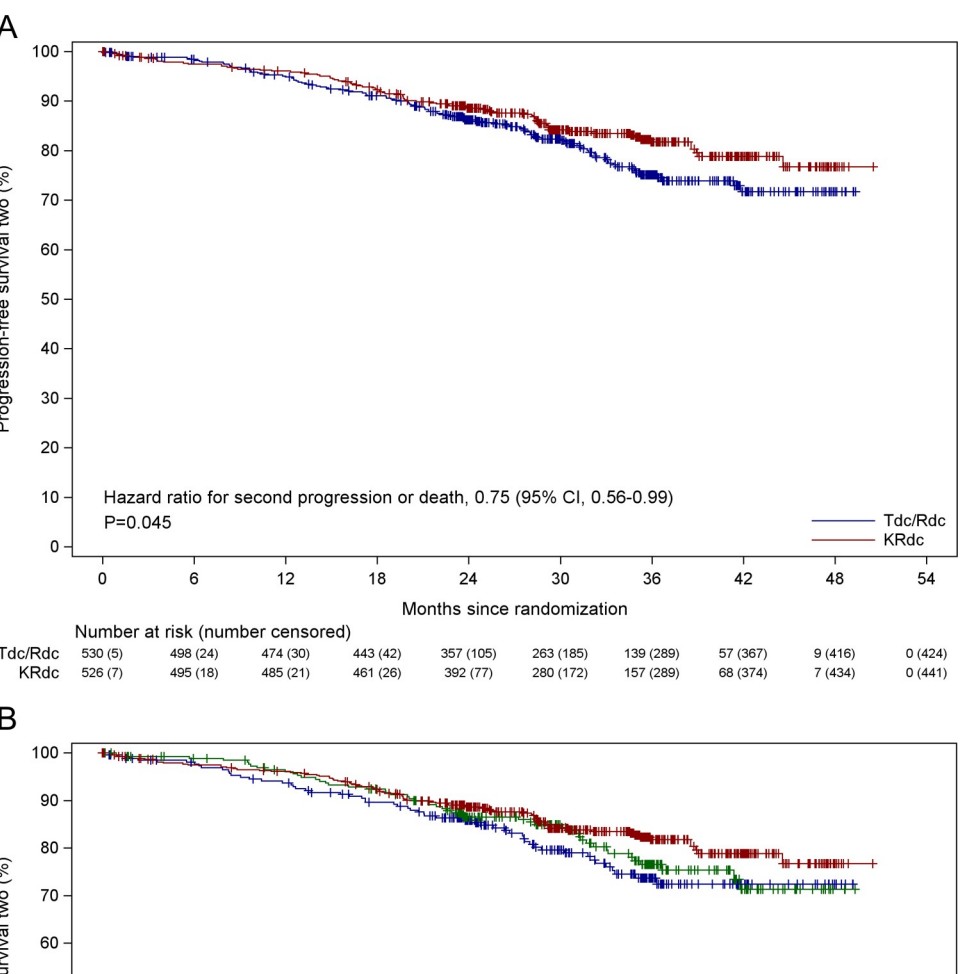

**Fig 3. Progression-free survival 2.** (A) Progression-free survival 2 for the KRdc group compared to the Tdc/Rdc control group. Note that medians and confidence intervals were inestimable. (B) Progression-free survival 2 for the KRdc group compared to the Tdc and Rdc control groups separately. Note that medians and confidence intervals were inestimable. KRdc, carfilzomib, lenalidomide, dexamethasone, and cyclophosphamide; Rdc, lenalidomide, dexamethasone, and cyclophosphamide; Tdc, thalidomide, dexamethasone, and cyclophosphamide.

peripheral sensory neuropathy from the addition of carfilzomib to Rdc. Grade 5 adverse events were reported during induction in 3 patients receiving KRdc and 4 patients receiving Tdc/Rdc control treatment. Thromboembolic events occurred at slightly higher rates in patients

receiving KRdc compared to Rdc. Thromboembolic events within the study have been examined in more detail in another paper [18]. The incidence of SAEs was 69.5% in those receiving KRdc and 55.3% in those receiving Tdc/Rdc control treatment. The majority of SAEs in all groups were due to infections or infestations.

Cardiac events were closely examined (Table 3). Grade 3 or 4 cardiac failure was reported in 4 patients (0.8%) in the KRdc group and no patients in the Tdc/Rdc control group, and pulmonary oedema in 2 (0.4%) patients in the KRdc group and no patients in the triplet control group. Grade 3 or 4 hypertension was reported in 2 patients (0.4%) in the KRdc group and 1 patient (0.2%) in the Tdc/Rdc control group.

### Exploratory analysis using optimal risk-adapted control group

To address whether the improvement in outcome with KRdc identified was attributable to the efficacy of the combination or was influenced by the control group that could, in the current treatment landscape, be considered suboptimal, we explored data from the risk-adapted randomisation step of the trial. Patients who achieved a suboptimal response (minimal or partial response) to triplet Tdc/Rdc induction were randomised between sequential triplet therapy with a proteasome inhibitor (CVD) or no further therapy prior to transplant. We have previously reported that CVD intensification was associated with improved outcome [9]. In order to remove this potential bias from the current analysis, we compared outcomes excluding those patients in the triplet control group who had achieved a partial or minimal response to initial induction and been randomised to the suboptimal approach of no further therapy prior to transplant. Results remained consistent with the intention-to-treat comparison (Fig C in S1 Text). The hazard ratio for the risk of progression or death was 0.64 (95% CI 0.52 to 0.78, $p <$ 0.001), corresponding to a 36% lower risk in the KRdc group compared to the optimal risk-adapted control group.

### Discussion

This is a large phase III study of a lenalidomide and carfilzomib combination regimen for newly diagnosed myeloma, with the results highlighting the efficacy and safety of this combination in previously untreated patients, supporting prior early phase studies [19,20]. The use of KRdc in this setting is associated with a significant improvement in PFS in comparison to a combination comprising either Tdc or Rdc, reducing the risk of progression or death by 37%. Recipients of KRdc induction were 4.35 times more likely to achieve at least a VGPR after a median of only 4 cycles of therapy, highlighting the rapidity of disease control. Responses of at least CR without bone marrow confirmation were seen in 56% with KRdc versus 34% with Tdc/Rdc pre-transplant, and 70% and 56% post-transplant, respectively, consistent with the benefit of the KRdc combination. Around 30% of cases had material available for assessment of MRD status; post-ASCT high rates of MRD negativity were attained, and importantly MRD − was associated with significantly longer PFS. This pattern of results, with a slightly higher rate of MRD− than of CR, is typical of myeloma studies reported recently; for example, IFM-2009 [21] reported CR rates of 59% and MRD− rates of 79% following ASCT. These differences reflect both the half-life of immunoglobulins and the availability of bone marrow confirmation of CR, a requirement of the definition.

Even within the group of patients achieving MRD negativity at a level of sensitivity of 0.004% (cutoff $4 \times 10^{-5}$ bone marrow leucocytes), there were differences in outcome, with the KRdc-treated group having significantly better PFS compared to those receiving Tdc/Rdc triplets. The cause for this difference is unknown, but we postulate there were deeper responses in the KRdc-treated group compared to the control group that translated to improved PFS.

**Table 3. Adverse reactions.**

| Adverse reaction | KRdc (n = 511) | | | | Rdc (n = 257) | | | | Tdc (n = 261) | | | |
|---|---|---|---|---|---|---|---|---|---|---|---|---|
| | Grade 1–2 | Grade 3 | Grade 4 | Grade 5 | Grade 1–2 | Grade 3 | Grade 4 | Grade 5 | Grade 1–2 | Grade 3 | Grade 4 | Grade 5 |
| **Haematological** | | | | | | | | | | | | |
| Anaemia | 354 (69.3%) | 51 (10.0%) | 1 (0.2%) | 0 | 185 (70.9%) | 14 (5.4%) | 1 (0.4%) | 0 | 174 (67.7%) | 12 (4.7%) | 0 | 0 |
| Neutrophil count decrease | 172 (33.7%) | 59 (11.5%) | 25 (4.9%) | 0 | 76 (29.1%) | 40 (15.3%) | 18 (6.9%) | 0 | 72 (28.0%) | 18 (7.0%) | 14 (5.4%) | 1 (0.4%) |
| Platelet count decrease | 213 (41.7%) | 27 (5.3%) | 16 (3.1%) | 0 | 80 (30.7%) | 6 (2.3%) | 0 | 0 | 35 (13.6%) | 3 (1.2%) | 0 | 0 |
| **Infections** | | | | | | | | | | | | |
| Cellulitis | 4 (0.8%) | 5 (1.0%) | 0 | 1 (0.2%)* | 1 (0.4%) | 1 (0.4%) | 0 | 0 | 0 | 1 (0.4%) | 0 | 0 |
| Lung infection | 31 (6.1%) | 50 (9.8%) | 3 (0.6%) | 1 (0.2%) | 25 (9.6%) | 18 (6.9%) | 1 (0.4%) | 0 | 18 (7.0%) | 20 (7.8%) | 2 (0.8%) | 1 (0.4%) |
| Sepsis | 1 (0.2%) | 10 (2.0%) | 14 (2.7%) | 1 (0.2%) | 0 | 7 (2.7%) | 3 (1.1%) | 2 (0.8%) | 0 | 5 (1.9%) | 2 (0.8%) | 0 |
| **Gastrointestinal** | | | | | | | | | | | | |
| Constipation | 208 (40.7%) | 1 (0.2%) | 0 | 0 | 114 (43.7%) | 2 (0.8%) | 0 | 0 | 158 (61.5%) | 1 (0.4%) | 1 (0.4%) | 0 |
| Diarrhoea | 136 (26.6%) | 15 (2.9%) | 0 | 0 | 64 (24.5%) | 4 (1.5%) | 0 | 0 | 50 (19.5%) | 5 (1.9%) | 0 | 0 |
| Nausea | 94 (18.4%) | 4 (0.8%) | 0 | 0 | 45 (17.2%) | 5 (1.9%) | 0 | 0 | 65 (25.3%) | 1 (0.4%) | 0 | 0 |
| Gastrointestinal—other | 3 (0.6%) | 2 (0.4%) | 0 | 0 | 1 (0.4%) | 0 | 0 | 1 (0.4%)* | 4 (1.6%) | 2 (0.8%) | 0 | 0 |
| **Neurological** | | | | | | | | | | | | |
| Peripheral motor neuropathy | 40 (7.8%) | 2 (0.4%) | 1 (0.2%) | 0 | 21 (8.0%) | 1 (0.4%) | 0 | 0 | 26 (10.1%) | 1 (0.4%) | 0 | 0 |
| Peripheral sensory neuropathy | 104 (20.4%) | 0 | 1 (0.2%) | 0 | 58 (22.2%) | 1 (0.4%) | 0 | 0 | 124 (48.2%) | 3 (1.2%) | 0 | 0 |
| Tremor | 29 (5.7%) | 0 | 0 | 0 | 30 (11.5%) | 1 (0.4%) | 0 | 0 | 58 (22.6%) | 0 | 0 | 0 |
| **Other** | | | | | | | | | | | | |
| Back pain | 47 (9.2%) | 4 (0.8%) | 0 | 0 | 41 (15.7%) | 2 (0.8%) | 0 | 0 | 32 (12.5%) | 7 (2.7%) | 0 | 0 |
| Cough | 53 (10.4%) | 7 (1.4%) | 0 | 0 | 17 (6.5%) | 2 (0.8%) | 0 | 0 | 23 (8.9%) | 2 (0.8%) | 0 | 0 |
| Dyspnoea | 71 (13.9%) | 5 (1.0%) | 0 | 1 (0.2%) | 18 (6.9%) | 7 (2.7%) | 0 | 0 | 40 (15.6%) | 4 (1.6%) | 0 | 0 |
| Fatigue/lethargy | 204 (39.9%) | 5 (1.0%) | 0 | 0 | 153 (58.6%) | 96 (36.8%) | 5 (1.9%) | 0 | 124 (48.2%) | 3 (1.2%) | 0 | 0 |
| Fever | 85 (16.6%) | 26 (5.1%) | 0 | 0 | 31 (11.9%) | 4 (1.5%) | 1 (0.4%) | 0 | 23 (8.9%) | 9 (3.5%) | 1 (0.4%) | 0 |
| Oedema limbs | 60 (11.7%) | 2 (0.4%) | 0 | 0 | 32 (12.3%) | 0 | 0 | 0 | 48 (18.7%) | 0 | 0 | 0 |
| Pain—other | 40 (7.8%) | 3 (0.6%) | 0 | 0 | 26 (10.0%) | 3 (1.1%) | 0 | 0 | 16 (6.2%) | 6 (2.3%) | 0 | 0 |
| Pulmonary embolism | 1 (0.2%) | 11 (2.2%) | 3 (0.6%) | 0 | 2 (0.8%) | 8 (3.1%) | 2 (0.8%) | 0 | 0 | 12 (4.7%) | 2 (0.8%) | 0 |
| Rash | 116 (22.7%) | 29 (5.7%) | 3 (0.6%) | 0 | 45 (17.2%) | 6 (2.3%) | 0 | 0 | 32 (12.5%) | 5 (1.9%) | 0 | 0 |
| **ARs of special interest** | | | | | | | | | | | | |
| Infusion reaction | 13 (2.5%) | 3 (0.6%) | 1 (0.2%) | 0 | 4 (1.5%) | 0 | 0 | 0 | 2 (0.8%) | 0 | 0 | 0 |
| Hypotension | 21 (4.1%) | 2 (0.4%) | 3 (0.6%) | 0 | 12 (4.6%) | 2 (0.8%) | 0 | 0 | 13 (5.1%) | 2 (0.8%) | 0 | 0 |
| Other thrombosis/embolism | 8 (1.6%) | 0 | 4 (0.8%) | 0 | 1 (0.4%) | 1 (0.4%) | 0 | 0 | 1 (0.4%) | 2 (0.8%) | 0 | 0 |
| Deep vein thrombosis | 30 (5.9%) | 6 (1.2%) | 0 | 0 | 10 (3.8%) | 2 (0.8%) | 1 (0.4%) | 0 | 17 (6.6%) | 4 (1.6%) | 0 | 0 |
| Heart failure | 1 (0.2%) | 4 (0.8%) | 0 | 0 | 0 | 0 | 0 | 0 | 0 | 0 | 0 | 0 |
| Hypertension | 1 (0.2%) | 2 (0.4%) | 0 | 0 | 2 (0.8%) | 0 | 0 | 0 | 0 | 1 (0.4%) | 0 | 0 |
| Pulmonary oedema | 2 (0.4%) | 2 (0.4%) | 0 | 0 | 0 | 0 | 0 | 0 | 0 | 0 | 0 | 0 |

*(Continued)*

**Table 3.** (Continued)

| Adverse reaction | KRdc (n = 511) | | | | Rdc (n = 257) | | | | Tdc (n = 261) | | | |
|---|---|---|---|---|---|---|---|---|---|---|---|---|
| | Grade 1–2 | Grade 3 | Grade 4 | Grade 5 | Grade 1–2 | Grade 3 | Grade 4 | Grade 5 | Grade 1–2 | Grade 3 | Grade 4 | Grade 5 |
| Myocardial infarction | 0 | 1 (0.2%) | 1 (0.2%) | 0 | 0 | 0 | 0 | 0 | 0 | 0 | 0 | 0 |
| Acute coronary syndrome | 1 (0.2%) | 0 | 0 | 0 | 0 | 0 | 0 | 0 | 0 | 0 | 0 | 0 |
| Cardiac disorders—other | 0 | 3 (0.6%) | 1 (0.2%) | 0 | 2 (0.8%) | 1 (0.4%) | 0 | 0 | 2 (0.8%) | 1 (0.4%) | 0 | 0 |

The safety population included all patients who received at least 1 dose of the trial treatment. Adverse reactions of any grade that were reported in >10% of patients in any treatment group, grade 3–5 in >5% of patients in any treatment group, or grade 5 in any treatment group are listed, along with other adverse events of special interest.

*Patients whose cause of death included "sepsis" in addition to the cause in this row.

AR, adverse reaction; KRdc, carfilzomib, lenalidomide, dexamethasone, and cyclophosphamide; Rdc, lenalidomide, dexamethasone, and cyclophosphamide; Tdc, thalidomide, dexamethasone, and cyclophosphamide.

We demonstrate that patients with high- and ultra-high-risk disease benefit from the use of KRdc, with no heterogeneity in the impact of the KRdc combination compared to standard-risk patients. This finding is encouraging, showing activity in patients with t(4;14) and del (17p), subgroups in whom there has been a lack of progress over the last decade [22]. Importantly, exposure to proteasome inhibitor and immunomodulatory agent combinations has been reported to be beneficial for these subgroups of patients [23], and potentially the greater activity of carfilzomib compared to bortezomib is associated with these excellent results. The superiority of carfilzomib compared to bortezomib was demonstrated in patients with relapsed disease in the ENDEAVOR study [5]. The superiority of carfilzomib is not, however, supported by data from the recently reported ENDURANCE study [24], which examined the combination of carfilzomib with lenalidomide and dexamethasone (without cyclophosphamide) as induction treatment for newly diagnosed myeloma patients. This study did not identify a difference in outcome between patients receiving KRd or bortezomib, lenalidomide, and dexamethasone (VRd), but it was not carried out in the same population as Myeloma XI+, excluding patients with high-risk disease and those destined for stem cell transplantation. For transplant-eligible patients of all risk subgroups, the phase II FORTE study [25] has reported preliminary data suggesting the KRd combination can achieve deeper responses than the combination of carfilzomib, cyclophosphamide, and dexamethasone, but full results are awaited.

An exploratory analysis demonstrated that KRdc was associated with prolonged PFS compared to both Rdc and Tdc when analysed separately. Any comparison between Rdc and Tdc is not powered within this contemporaneously recruited cohort, but from the preceding Myeloma XI trial, Rdc was associated with prolonged PFS, PFS2, and OS compared to Tdc [8]. KRdc was associated with significantly prolonged PFS2 compared to the pre-planned Tdc/Rdc triplet control group comparator. On exploratory analysis, KRdc was associated with prolonged PFS2 compared with Tdc, but not Rdc. However, the follow-up remains immature for this endpoint, which will be re-examined, along with OS, at final analysis.

The KRdc combination was well tolerated in the present study, with a low incidence of peripheral neuropathy. Interestingly, there was also less neutropenia in the KRdc group, possibly due to the fewer cycles required to achieve maximum response (median 4 versus 5 cycles) and the associated rapid, deep responses with improved bone marrow function. In relapsed patients in the ENDEAVOR study [5,26], grade 3 or higher cardiac failure were seen in 22/462 (4.8%) carfilzomib-treated patients compared to only 8/456 (1.8%) patients in the control group. In the newly diagnosed myeloma setting investigated here, the overall incidence of cardiac adverse events was low, which may be related to the different dose of carfilzomib used in

our study (36 mg/m$^2$ versus 56 mg/m$^2$) or the absence of the impact of prior therapy. The exclusion of patients with a significant history of cardiac disease was at the discretion of the investigators and so may have affected this finding. The treatment was delivered in community as well as academic hospitals, with no significant difficulties encountered.

A potential criticism of the Myeloma XI+ study relates to the control arm, where a triplet combination that was a standard of care in prior UK trials and that is in widespread use across the world was used. A comparison of a different immunomodulatory drug and proteasome inhibitor combination (e.g., VRd or bortezomib, thalidomide, and dexamethasone) would also have been of interest because of more recent uptake of such combinations in the EU and US, but this was not the position when the study was initially implemented. Despite this, the outcome for the control arm in this study compares well to other combinations, with a median PFS of 36.2 months. In the IFM 2009 study [21], VRd was associated with a median PFS of 36 versus 50 months for those who underwent ASCT. In the IFM study all patients received lenalidomide maintenance, in contrast to patients in the Myeloma XI+ study, where only 50% received maintenance. In real-world data reported from the Mayo Clinic [27], which included 243 patients who received VRd followed by ASCT, half of whom received maintenance therapy, VRd was associated with a median PFS of 28 months. To further address this question we utilised data from the current trial, where an immunomodulatory-agent triplet was used initially followed by a proteasome triplet for suboptimal responders not achieving at least a VGPR. Using these data we were able to generate optimised outcome data, but even in comparison to this, the use of KRdc was associated with significantly improved outcomes. Another limitation includes the unblinded nature of the therapy randomisation; both the patient and local investigator were aware of the treatment being delivered.

In conclusion, in transplant-eligible, newly diagnosed multiple myeloma patients in the Myeloma XI+ study, a carfilzomib and lenalidomide combination, KRdc, was well tolerated and was associated with an increased percentage of patients achieving at least a VGPR, more MRD-negative responses, and significantly prolonged PFS compared to a immunomodulatory-agent-based triplet induction combination.

## Supporting information

**S1 Text.** Fig A—Progression-free survival by cytogenetic risk. Progression-free survival (PFS) for carfilzomib, lenalidomide, dexamethasone, and cyclophosphamide (KRdc) compared to the triplet control group (Rdc/Tdc) within each cytogenetic risk group. (A) Standard risk, (B) high risk, (C) ultra-high risk. Adverse molecular abnormalities were defined as gain(1q), del (17p), t(4;14), t(14;16), or t(14;20). Efficacy in the subgroups of standard risk, high risk (defined as 1 adverse cytogenetic abnormality) and ultra-high risk (2 or more adverse cytogenetic abnormalities) were pre-specified by protocol. CI, confidence interval; HR, hazard ratio; m, months. Fig B—Progression-free survival for patients with/without t(4;14) and del (17p). Progression-free survival (PFS) for carfilzomib, lenalidomide, dexamethasone, and cyclophosphamide (KRdc) compared to the triplet control group (Rdc/Tdc) for patients with the cytogenetic lesions t(4;14) and del(17p). (A) t(4;14), (B) no t(4;14), (C) del(17p), (D) no del (17p). CI, confidence interval; HR, hazard ratio; m, months. Fig C—Progression-free survival adjusted for CVD randomisation. Progression-free survival (PFS) for carfilzomib, lenalidomide, dexamethasone, and cyclophosphamide (KRdc) compared to the triplet control group (Rdc/Tdc) with patients achieving a suboptimal response to triplet treatment and randomised to no intensification therapy removed. CI, confidence interval; HR, hazard ratio; m, months. Table A—Myeloma XI+ study sites and principal investigators. Text A—CONSORT Checklist.

Text B—Myeloma XI+ protocol.
(PDF)

## Acknowledgments

We thank all the patients at centres throughout the UK whose willingness to participate made this study possible. We are grateful to the UK National Cancer Research Institute Haemato-oncology Clinical Studies Group, UK Myeloma Research Alliance, and all principal investigators, sub-investigators, and local centre staff for their dedication and commitment to recruiting patients to the study. We thank the members of the Myeloma XI+ Trial Steering Committee and Data Monitoring and Ethics Committee. The support of the Clinical Trials Research Unit at the University of Leeds was essential to the successful running of the study; we thank all its staff who have contributed, past and present. Central laboratory analysis was performed at the Institute of Immunology and Immunotherapy, University of Birmingham; the Institute of Cancer Research, London; and the Haematological Malignancy Diagnostic Service, St James's University Hospital, Leeds. We are very grateful to the laboratory teams for their contribution to the study. We acknowledge support from the National Institute for Health Research Biomedical Research Centre at the Royal Marsden Hospital and the Institute of Cancer Research.

## Author Contributions

**Conceptualization:** Graham H. Jackson, Charlotte Pawlyn, David A. Cairns, Nigel H. Russell, Mark T. Drayson, Walter M. Gregory, Roger G. Owen, Faith E. Davies, Gareth J. Morgan.

**Data curation:** Graham H. Jackson, Charlotte Pawlyn, David A. Cairns, Mark T. Drayson, Martin F. Kaiser, Roger G. Owen.

**Formal analysis:** Graham H. Jackson, Charlotte Pawlyn, David A. Cairns, Walter M. Gregory.

**Funding acquisition:** Graham H. Jackson, Nigel H. Russell, Walter M. Gregory, Faith E. Davies, Gareth J. Morgan.

**Investigation:** Graham H. Jackson, Charlotte Pawlyn, David A. Cairns, Ruth M. de Tute, Anna Hockaday, Corinne Collett, John R. Jones, Bhuvan Kishore, Mamta Garg, Cathy D. Williams, Kamaraj Karunanithi, Jindriska Lindsay, Alberto Rocci, John A. Snowden, Matthew W. Jenner, Gordon Cook, Mark T. Drayson, Walter M. Gregory, Martin F. Kaiser, Roger G. Owen, Faith E. Davies, Gareth J. Morgan.

**Project administration:** Graham H. Jackson, David A. Cairns, Anna Hockaday, Corinne Collett, Walter M. Gregory.

**Resources:** Graham H. Jackson, Charlotte Pawlyn, David A. Cairns, Ruth M. de Tute, Bhuvan Kishore, Mamta Garg, Cathy D. Williams, Kamaraj Karunanithi, Jindriska Lindsay, Alberto Rocci, John A. Snowden, Matthew W. Jenner, Gordon Cook, Nigel H. Russell, Mark T. Drayson, Walter M. Gregory, Martin F. Kaiser, Roger G. Owen, Faith E. Davies, Gareth J. Morgan.

**Supervision:** Graham H. Jackson.

**Visualization:** Graham H. Jackson, Charlotte Pawlyn, David A. Cairns, John R. Jones.

**Writing – original draft:** Graham H. Jackson, Charlotte Pawlyn, David A. Cairns, Faith E. Davies, Gareth J. Morgan.

**Writing – review & editing:** Graham H. Jackson, Charlotte Pawlyn, David A. Cairns, Ruth M. de Tute, Anna Hockaday, Corinne Collett, John R. Jones, Bhuvan Kishore, Mamta Garg, Cathy D. Williams, Kamaraj Karunanithi, Jindriska Lindsay, Alberto Rocci, John A. Snowden, Matthew W. Jenner, Gordon Cook, Nigel H. Russell, Mark T. Drayson, Walter M. Gregory, Martin F. Kaiser, Roger G. Owen, Faith E. Davies, Gareth J. Morgan.

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
