## [Editor Report · Decision Letter 0]

4 Mar 2020

Dear Dr Jackson, 

Thank you for submitting your manuscript entitled "Carfilzomib, Lenalidomide, Dexamethasone and Cyclophosphamide (KRdc) as Induction Therapy for Transplant Eligible Newly Diagnosed Multiple Myeloma." for consideration by PLOS Medicine.

Your manuscript has now been evaluated by the PLOS Medicine editorial staff and I am writing to let you know that we would like to send your submission out for external peer review.

Kind regards,

Richard Turner, PhD

Senior editor, PLOS Medicine

rturner@plos.org

---

## [Decision Letter · Decision Letter 1]

7 Aug 2020

Dear Dr. Jackson,

Thank you very much for submitting your manuscript "Carfilzomib, Lenalidomide, Dexamethasone and Cyclophosphamide (KRdc) as Induction Therapy for Transplant Eligible Newly Diagnosed Multiple Myeloma." (PMEDICINE-D-20-00730R1) for consideration at PLOS Medicine. We do apologize for the delay in sending you a decision. 

Your paper was evaluated by the editors and sent to independent reviewers, including a statistical reviewer. The reviews are appended at the bottom of this email and any accompanying reviewer attachments can be seen via the link below:

[LINK]

In light of these reviews, we will not be able to accept the manuscript for publication in the journal in its current form, but we would like to invite you to submit a revised version that addresses the reviewers' and editors' comments fully. You will appreciate that we cannot make a decision about publication until we have seen the revised manuscript and your response, and we expect to seek re-review by one or more of the reviewers. 

We hope to receive your revised manuscript by Aug 27 2020 11:59PM. Please email us (plosmedicine@plos.org) if you have any questions or concerns.

Please let me know if you have any questions. Otherwise, we look forward to receiving your revised manuscript soon. 

Sincerely,

Richard Turner, PhD

rturner@plos.org

Please review PLOS Medicine's data policy (https://journals.plos.org/plosmedicine/s/data-availability) which your paper will need to comply with in the event of publication. For example, a non-author contact would need to be provided for those inquiring about access to study data. 

Please add a study descriptor to the title, e.g., "...myeloma: an open-label randomized controlled trial".

Please combine the "methods" and "findings" subsections of your abstract to match journal style. The final sentence of the new combined subsection should quote 2-3 of the study's main limitations. 

Please add summary demographic details for study participants to the abstract. 

After the abstract, we will need to ask you to add a new and accessible "author summary" section in non-identical prose. You may find it helpful to consult one or two recent research papers in PLOS Medicine to get a sense of the preferred style. 

Please remove the "role of the funding source" section from the text. This information will appear in the article metadata in the event of publication, via entries in the submission form. Similarly, please remove the conflict of interest information from the end of the main text. 

Please avoid claims of "the first" (e.g., at the start of the discussion section) and where necessary add "to our knowledge" or similar. 

Please restructure the discussion section so that there is a discrete paragraph summarizing study limitations. 

Should that be "these data" on p.14?

Please substitute "sex" for "gender" where appropriate, e.g., in figure 1b.

Throughout the paper, please quote exact p values or p<0.001.

Please aim to be consistent about phase 3/III; and US/UK spelling (e.g., "haematological"/"multi-center").

Throughout the text, please adapt reference call-outs as follows: "... are published [8,9]." (i.e. preceding punctuation and with no spaces within the square brackets). 

In the reference list, please ensure that journal names are abbreviated consistently; and where appropriate list 6 author names rather than 3, followed by "et al.".

Please move the patient flowchart to the main text. 

Please add a completed CONSORT checklist as an attached file, referred to in the methods section of the main text. In the checklist, individual items should be referred to by section (e.g., "Methods") and paragraph number rather than by line or page numbers, as the latter generally change in the event of publication. 

Comments from the reviewers:

*** Reviewer #1: 

Alex McConnachie, Statistical Review

This is a review of the statistical aspects of Jackson et al's paper about the interim results of the Myeloma XI+ trial, comparing progression free survival between patients treated with a 4-drug combination versus one of two 3-drug combinations.

This is a hugely impressive trial, and I am in favour of the decision to seek to publish the results of the study to date. The statistical analyses are generally very good, and my comments are mainly to do with the way that things have been presented.

The co-primary endpoints are PFS and overall survival, though this paper does not report the OS data. This is fine, but reading the abstract, the reader is expecting to see them. The results in the abstract show PFS results, and then moves on to response rates and adverse events, giving the impression that the OS data are being omitted. I think the first part of the abstract, and maybe the paper as a whole (including the title?), could be more up-front about this being a planned interim analysis, focusing on the first of two co-primary outcomes, plus secondary and safety outcomes. This is clearer on reading the statistical methods section, but many readers will not get that far.

The wording in the abstract "…median PFS not reached vs 36.2 months" is not immediately clear. Perhaps a few more words need to be used to explain this a little better, for someone who is only reading the abstract.

The unblinded nature of the trial is not mentioned as a limitation. Should it be?

Particularly as a non-clinical reader, I found the use of abbreviations quite challenging at times throughout the paper. Often they were not spelled out in full on first use, making it difficult to follow.

The primary analysis method of an adjusted Cox model is fine. I presume the proportional hazards assumption was checked, and was met, though this is not mentioned.

A subgroup analysis is presented of the main treatment effect within those who were MRD negative (though I had to look up what MRD means). Is this a sensible analysis, given that MRD status is defined after randomisation (and treatment)? Should the same results from the MRD+ group be reported? Or, should the main analysis be performed with adjustment for MRD status, to assess whether MRD status explains the treatment effect?

The final efficacy result reported is that KRdc showed improved PSF2 compared to CTD, but not CRD, when analysed separately. Given that the KRdc vs. CTD/CRD comparison only just achieved p<0.05, it is almost certain that at least one of KRdc vs. CTD or CRD will not achieve this p-value threshold. Is this observation worthy of a mention, when I doubt that there is any evidence of a difference between CRT and CRD?

Table 2 present response data, but does not show any treatment effect estimates or p-values. These might be helpful here.

Figures 2a and 2b include legends saying that none of the median PSF2 times, nor their confidence limits, were estimable. Is this necessary?

*** Reviewer #2: 

The manuscript by Jackson et al describes the first interim analysis of one of the randomisations on the UK Myeloma XI trial. The randomisation compared KRdc against CRD and CTD (2:1:1) with the latter two arms also undergoing a second randomisation in the event of suboptimal response. The design is thus complex with numerous caveats in interpretation of the results and the control arms (CTD or CRD) are not considered standard induction in most countries where proteasome inhibitor-based induction (+/- IMiD) is routine. Nonetheless, this is an important study and needs to be published to a wide audience. The PFS benefit for the KRdc arm is not only statistically significant but is clinically significant as well. The results are likely to be of interest to readers of Plos Medicine.

Major points

* Methods (p6). It is hopefully a typo, but PFS is defined as the time from maintenance randomisation until progression or death. Please confirm if this definition is correct (rather than from induction randomisation). If it is correct, then results really need to be reanalysed with PFS defined according to correct definition. Same as per OS - please define accurately in view of above point.

* Methods (p7). Is it valid to combine CTD and CRD into one comparative group? In describing the primary endpoint analysis, the Protocol (Section 15.3) states that "To compare RCD with CTD, CCRD with RCD/CTD combined or with whichever of these two is superior" which is not exactly what is communicated in the Methods section. In the Study Design and Participant section of this paper it states that PFS and OS were significantly better for CRD compared to CTD. In the Results section an exploratory analysis of PFS in KRdc compared to both CRD and CTD induction when analysed separately is presented but this analysis is not mentioned or defined in the statistical section. Ultimately that data appears reasonable as both analysis types are presented for readers but the rationale to justify why the CRD and CTD arms can be compared into one group as opposed to performing comparisons of three separate arms, and the analysis method for the three group comparison needs to be tidied up in the statistical methods section.

Minor points

* Methods (p5). Why were calcium and platelets chosen as minimisation factors in the randomisation as opposed to albumin which is a key prognostic variable in the ISS? Was the lased on some prior findings from the UK Myeloma group?

* Methods (p5). Is it valid to compare treatment arms with different dose densities and cycle numbers? The dexamethasone dose in the CTD is greater because of the shortened cycle length and the CTD and CTD arms have more cycles, presumably due to worse response rates. This results in treatment arms that are not directly comparable. Nonetheless, at least in terms of response and PFS efficacy, this would bias against the KRdc.

* Methods (p6). What was the definition of renal insufficiency for using MEL200? Please list whether eGRF cut-off or up to investigator discretion

* Methods (p7). Which group of patients had MRD assessments performed? Only patients in CR? All patients?

* Methods (p7). What methods were used for serum FLC and urine light chain excretion?

* Methods (p7). Why were AEs not assessed in the CTD group?

* Results (Supplementary Fig 1). In the consort diagram it is not clear what has happened at various points as the numbers do not add up e.g. 511pts received KRdc but only 474 had response recorded. What happened to the others e.g. death, off study for toxicity etc. Similarly, it would be helpful to have the broad reasons for not proceeding to ASCT listed in this Figure.

* Results (p9). Patients older than 65 had a greater PFS benefit with KRdc. This is somewhat unexpected. Can the authors speculate in the discussion why this might be the case?

* Results. While OS data is not mature, can the authors please provide the number of deaths in each arm.

* Results (p12). The cardiac event rate in the KRdc arm is very low. It seems that exclusion of patients with a history of cardiac disease was largely at investigator discretion. Perhaps this could be mentioned either in the Methods or Discussion as there will be some uncertainty about the extent of exclusion of patients with a history of cardiac disease.

* Results (Table 1). It appears that molecular risk was only known for ~ 35% of patients which is disappointing. It makes the conclusions about the benefit of KRdc in high and ultra-high risk myeloma harder to support e.g. in Supplementary Fig 2, while HRs are consistent with primary analysis, there is in fact no significant difference between arms for any of the cytogenetic risk groups, particularly not the ultra-high-risk group. While this is probably due to small numbers, it could also be because there is no beneficial effect of KRdc in the ultra-high-risk group. In a similar fashion, the conclusion states that there is promising activity in patients with t(4;14) and 17p- patients but no data in the Supplementary appendix is provided to support this (the high-risk curved in Suppl Fig 2 are overwhelmed by 1q patients). Could data on the t(4;14) and 17p- groups be presented in the Supplementary appendix, acknowledging that numbers will be very small?

* Results (Table 2). The post-ASCT response rates appear to be per-protocol and not IIT. Could this be clearly indicated?

*** Reviewer #3: 

I congratulate the authors on their thoughtful study design and well-written manuscript. Unfortunately the study's impact will be limited primarily by the inclusion of CTD-treated patients in the control arm, as thalidomide has largely become antiquated due to its inferior efficacy and tolerability issues and is now only rarely used, at least in the United States. Of course this is not the fault of the authors as the study was initiated prior to the de facto retirement of thalidomide. What I found most thought provoking about this study was the signal of high efficacy in the high-risk subgroup. Although the point of the study was to evaluate the value of adding carfilzomib to an IMiD-Cytoxan triplet, the more relevant question that the study raises in my opinion is the value of adding Cytoxan to an IMiD-PI triplet (or even, very ambitiously and probably unwisely, to an IMiD-PI-mAb quad) in patients with high-risk NDMM. The EVOLUTION study attempted to answer the question regarding the value of an IMiD-PI-Cytoxan quad vs triplets and failed to demonstrate a benefit of adding Cytoxan to VRD, however the overall number of subjects was low to the point of precluding any meaningful subgroup analysis. Certainly the Myeloma XI+ high-risk subgroup analysis is not statistically significant but the HR in the 17p- subgroup for example is enough to give one pause.

It is also noteworthy that the high-grade hematologic toxicity in the KRDc arm was lower than what I would have expected. The EVOLUTION investigators chose a Rev dose of 15mg for this reason, thinking that 25mg would be too toxic hematologically in a VRDc quad, and it seems that Myeloma XI+ may have proven otherwise, albeit with a relatively few number of induction cycles.

Although this was not a study of a combined IMiD-PI vs planned sequential IMiD->PI induction approach, the results nonetheless confirm the power of IMiD-PI synergy and suggest that a response-adapted approach that separates IMiD from PI and attempts to deepen response to the former with a switch to the latter is likely not an effective strategy.

Accordingly, I do think that the Myeloma XI+ trial is worthy of publication in this journal as it will make a meaningful contribution to the existing body of knowledge in the NDMM space. I think the manuscript is high-quality overall but can be improved further:

1) "This is the first phase III randomised study of a carfilzomib and lenalidomide combination induction therapy for TE NDMM patients." Is this completely accurate? What about ENDURANCE and FORTE?

2) Page 11 paragraph 2: The inclusion of reasons for early stoppage of induction therapy in both arms is appreciated. I am assuming that the definition of "early" is before completion of the fourth cycle. I am curious as to the reasons for stopping induction therapy in general (not just stopping therapy early) in both arms. Since median # of induction cycles was two cycles fewer in the KRDc arm and the ≥VGPR rate was higher in this arm, it is most likely that the latter is the explanation for the former, in other words the quad-treated subjects achieved maximal response with fewer cycles compared to those treated with a triplet. However it is also possible that some subjects may have stopped KRDc after the fourth cycle due to toxicity and prior to the achievement of maximal response (aka prior to M-spike plateau in patients who hadn't yet achieved CR). In order for this study to retain maximal significance in the current era (and specifically thinking about the coming IMiD-PI-mAb quad induction era), it would be nice to have a better idea of how tolerable KRDc is beyond four cycles.

3) Page 11 paragraph 3: "Thromboembolic events occurred in similar proportions of patients across the treatment groups." This statement is probably true vis a vis the KRDc arm (9.4% DVT+other thrombosis) vs CRD/CTD control arm (7.5%), but probably not in terms of the KRDc arm vs the CRD arm specifically (5.8%). We know that Thal is quite thrombogenic, and based on data such as the recently presented MSKCC data (Piedra et al, ASH 2019 abstract #1835) we now know that KRD is more thrombogenic than RVD (VTE rate 4x higher in patients on ASA prophylaxis!). In my opinion, the statement of "similar proportions" may give the reader the impression that carfilzomib does not add thrombotic risk to a Revlimid-based regimen when we have a pretty good idea that it does and when the Myeloma XI+ data describe an almost 10% incidence of thrombosis with KRDc. In my opinion, the portion of this manuscript that remains most relevant today is the KRDc vs CRD analysis, which the authors do a nice job of in terms of efficacy, but I would kindly suggest maybe something similarly robust in terms of toxicity as well.

4) Page 13 paragraph 2: If the "potentially greater activity of carfizomib compared to bortezomib" is to be mentioned, I think there probably should be at least a brief discussion of ENDURANCE in the discussion section.

Congratulations again on an overall excellent manuscript.

***

[LINK]

---

## [Decision Letter · Decision Letter 2]

26 Oct 2020

Dear Dr. Jackson,

Thank you very much for re-submitting your manuscript "Carfilzomib, Lenalidomide, Dexamethasone and Cyclophosphamide (KRdc) as Induction Therapy for Transplant Eligible Newly Diagnosed Multiple Myeloma Patients: Myeloma XI+, an open-label randomised controlled trial." (PMEDICINE-D-20-00730R2) for consideration at PLOS Medicine.

I have discussed the paper with editorial colleagues and it was also seen again by two reviewers. I am pleased to tell you that, provided the remaining editorial and production issues are fully dealt with, we expect to be able to accept the paper for publication in the journal.

[LINK]

Please let me know if you have any questions. Otherwise, we look forward to receiving the revised manuscript shortly.

Sincerely,

Richard Turner, PhD

rturner@plos.org

Requests from Editors:

So as to comply with PLOS Medicine's data policy, please include de-identified patient data for the current analysis, either in the form of supplementary files or at a publicly-accessible repository. 

Please remove "methodologically sound" from your data statement. 

Please adapt your title so that "(Myeloma XI+)" appears immediately before the colon.

In the "Background" subsection of your abstract, please add a few words to indicate the clinical indication(s) in which carfilzomib is efficacious.

Please mention in the abstract where the trial was done. 

Please quote dates of start and end of participant recruitment in the abstract. 

In the "Conclusions" subsection of your abstract and other relevant points in the text, please rephrase "deeper responses", adding a few words to make the meaning clear. 

The introduction section of the main text is currently very short, and we suggest adding a few additional sentences of introduction about the disease in question and current treatment options. 

In the first paragraph of your discussion section and at the end of the main text, please adapt the phrasing to note that the combination therapy "was" (well tolerated in the present study). 

Throughout the text, please remove spaces from within the square brackets containing reference call-outs (e.g., " ... early phase studies [19,20]."). 

Please remove the information on data sharing from the end of the main text. This information will appear in the article metadata in the event of publication, via entries in the submission form. 

In your reference list, please ensure that journal names are abbreviated consistently (e.g., "N Engl J Med."). 

We were unable to find a completed CONSORT checklist with your submission - please include this as a supplementary document with your resubmission, referred to in the methods section of your main text (e.g., "See S1_CONSORT_Checklist"). 

In the completed checklist, please ensure that individual items are referred to by section (e.g., "Methods") and paragraph number rather than by page or line numbers, as the latter generally change in the event of publication. 

Similarly, we were unable to find the trial protocol. Please include this as a supplementary document with your resubmission, referred to in the methods section. 

Comments from Reviewers:

*** Reviewer #1: 

Alex McConnachie, Statistical Review

I thank the authors for their consideration of my original points. I am satisfied with their responses.

I understand the desire to not replicate results between the text and the tables, but I still prefer to see treatment effect estimates presented systematically in tabular form (i.e. added to Table 2). However, this is not a critical point.

*** Reviewer #2: 

Thank you for addressing the comments. I have no further suggestions. I think the manuscript reads well.

***

[LINK]

---

## [Editor Report · Decision Letter 3]

23 Nov 2020

Dear Prof Jackson, 

On behalf of my colleagues and the academic editor, Dr. Peter N Mollee, I am delighted to inform you that your manuscript entitled "Carfilzomib, Lenalidomide, Dexamethasone and Cyclophosphamide (KRdc) as Induction Therapy for Transplant Eligible Newly Diagnosed Multiple Myeloma Patients (Myeloma XI+): interim analysis of an open-label randomised controlled trial." (PMEDICINE-D-20-00730R3) has been accepted for publication in PLOS Medicine. 

PRODUCTION PROCESS

Before publication you will see the copyedited word document (within 5 business days) and a PDF proof shortly after that. The copyeditor will be in touch shortly before sending you the copyedited Word document. We will make some revisions at copyediting stage to conform to our general style, and for clarification. When you receive this version you should check and revise it very carefully, including figures, tables, references, and supporting information, because corrections at the next stage (proofs) will be strictly limited to (1) errors in author names or affiliations, (2) errors of scientific fact that would cause misunderstandings to readers, and (3) printer's (introduced) errors. Please return the copyedited file within 2 business days in order to ensure timely delivery of the PDF proof. 

If you are likely to be away when either this document or the proof is sent, please ensure we have contact information of a second person, as we will need you to respond quickly at each point. Given the disruptions resulting from the ongoing COVID-19 pandemic, there may be delays in the production process. We apologise in advance for any inconvenience caused and will do our best to minimize impact as far as possible.

EARLY VERSION

PRESS

PROFILE INFORMATION

Thank you again for submitting the manuscript to PLOS Medicine. We look forward to publishing it. 

Best wishes, 

Richard Turner, PhD

Senior Editor 

PLOS Medicine

plosmedicine.org